# Consumer Awareness of the Regional Food Market: The Case of Eastern European Border Regions

**DOI:** 10.3390/foods8100467

**Published:** 2019-10-10

**Authors:** Andrzej Soroka, Julia Wojciechowska-Solis

**Affiliations:** 1Department of Medical Sciences and Health Sciences, Siedlce University of Natural Sciences and Humanities, 08110 Siedlce, Poland; wachmistrz_soroka@o2.pl; 2Department of Agrobioengineering, University of Life Sciences in Lublin, 20950 Lublin, Poland

**Keywords:** food choice, consumer behavior, consumer competence, regional food products, Eastern Poland, Western Ukraine

## Abstract

The aim of this paper is to determine the awareness measures of consumers from Eastern Poland and Western Ukraine towards regional food products, including consumer knowledge on regional products available in the media and their availability on the food market. The effort was made to compare consumers’ opinions on the reasons for purchasing regional food and ways of distinguishing it from conventional products, as well as on the availability of regional products. Consumer awareness—that is, making informed choices based on the knowledge we have—is a measure of attitudes and cognition, and sometimes can be directed towards the brand, which is the product’s regional designation. Therefore, it is necessary to comment that attitudes towards regionality can generate a behavioral intent. A diagnostic survey with an author’s questionnaire was used in the study, which helped to survey 1128 respondents from Eastern Poland—that is, from the Podlaskie, Lublin, and Subcarpathia regions—and 1072 from Western Ukraine, including the Volyn, Lviv, and Transcarpathia regions. Discriminant function analysis was used in statistical analysis. Both residents of Eastern Poland and Western Ukraine obtained information on regional food products from their friends or family and from television (TV), internet, and regional fairs. Consumers from both countries pointed at too many possibilities of purchasing regional products; at the same time, they paid attention to a limited number of points of sale. TV and Internet have a great promotional potential to educate young consumers focused on the purchase of regional food products.

## 1. Introduction

Globalization of the food market has caused the intensification of food control in terms of its origin [1], quality [2], health, and sustainability. The food industry takes a broad view of the term “food control”, which includes a large number of factors, such as [3]:-safety: setting standards for toxicological and microbiological hazards, and instituting procedures and practices to ensure that the standards are achieved;-nutrition: maintaining nutrient levels in food ingredients, and formulating foods with nutritional profiles that contribute to consumer interest in healthful diets;-quality: providing sensory characteristics, such as taste, aroma, palatability, and appearance;-value: providing characteristics of consumer utility and economic advantage, involving attributes such as convenience, packaging, and shelf life.

The main reason for food control is the growing number of consumers and producers in the food chain [4]. In many countries, there is support for local food production as an alternative system to conventional food [5]. Additionally, there has been an increase in the importance of local food supply chains, which are considered to be relatively sustainable [6]. Purchasing local goods can be motivated by environmental, social, economic, and especially health aspects [7,8,9,10].

Agri-food producers are constantly looking for ways to help consumers identify the quality of products and clarify issues related to food security and environmental protection, since contemporary consumers are willing to pay a higher price for a high-quality product with a well-known source of origin [11,12,13]. 

Tastes and preferences of society, as well as the authenticity of products, play key roles in food production [14]. The studies conducted in Germany by Heer and Mann [15] point at to the effectiveness of creating regional food supply chains. Most of all, they bring economic benefits to entrepreneurs from the local community [16], who are able to function in the niche market of regional products [17]. Regional food supply chains also affect elements such as information, the nature of the regional product, media, advertising, and other behavioral factors. In the context of regional/traditional food consumption, the role of awareness measures is the most important in the study of consumer behavior [18].

The interest in regional products is systematically growing, not only in European countries, but also in the USA and Japan. For instance, in Japan, the local food system called the “Chisan-Chisho Movement” (“Locally Produced, Locally Consumed”). It is promoted in the context of sustainable food production [19], whereas in the United States the following programs are popular: the “Local food purchase policy” [20] and the “AgriMissouri Promotion Program” [21]. In the United States, a number of tools for objectively measuring the local food environment have been developed and validated. The Farmers’ Market audit tool and a new tool to measure community members’ perceptions of the local food environment have also been released (“From Farm to Table”) [22]. In the United Kingdom, the demand for regional products continues to grow by an average of 6% per year. The European Union regulations create favorable conditions for the development of production of local products and their sale in fairs and local production shops, as well as public food procurement [14].

The aim of this study is to identify awareness measures of consumers of Eastern Poland and Western Ukraine towards regional food products. The effort was made to determine the reasons for buying regional food and ways to distinguish it from conventional products. The availability of purchasing regional products was also assessed. The aim was also to determine the sources from which the consumer derives information about regional products. Two regions were selected on purpose. There is no barrier to population movement, despite the functioning of the border. Poland is a member of the European Union (UE), while Ukraine has applied for EU membership. Despite the proximity of these two regions, there are significant differences in the functioning of their food markets, as well as differences in consumer awareness [23]. 

## 2. Materials and Methods 

### 2.1. Applied Research Methods

A diagnostic survey, aimed at discovering the reasons for the current state of consumer awareness measures, along with the authors’ questionnaire was used in the studies. Five out of total 11 questions asked in the questionnaire were analyzed in this paper. The survey was prepared in two languages: a survey in Polish for the research area in Eastern Poland, and a survey in Ukrainian for the research area in Western Ukraine. The study was conducted in May and June 2018. Validation is required for questionnaires developed by the researchers on an ad-hoc basis for the needs of a specific study and, as new measuring tools, these questionnaires are of unknown quality. The results discussed in this paper were obtained from a new questionnaire developed by the authors—therefore, the validation procedure was used [24]. After the use of construction and validation procedures, a five-point Likert scale was applied to measure consumer awareness (where 1 = low importance for consumer and 5 = high importance for consumer). An indicator of scale reliability was calculated where Cronbach’s α was 0.85. In the sample selection procedure, a stratified random sampling was applied for each group of respondents, separately for the consumers of Eastern Poland and those from Western Ukraine. In the next stage, respondents were proportionally divided, taking into account their gender and place of residence, at which point the consumers from rural and urban areas were identified. Methodological procedures allowed calculating the size of the sample, where the level of confidence was set at 0.95, the estimated size fraction at 0.50, and the maximum error at 0.05. After taking into account the gender and place of residence of the population under survey, a quota selection was used, where respondents were selected on the basis of their availability. 

### 2.2. Selection of the Research Sample

A diagnostic survey with the authors’ questionnaire was used in the study to examine 1128 respondents from the Podlaskie, Lublin, and Subcarpathia regions of Eastern Poland, as well as 1072 from the Volyn, Lviv, and Transcarpathia regions of Western Ukraine. There were 1200 respondents surveyed from each country. The sample were comprised of individuals who reported purchasing regional products. Participation was voluntary, and all of them completed the questionnaire. Some of the questionnaires were rejected due to lack of reliability and errors. The survey was conducted by telephone (using a computer-assisted telephone interviewing technique), and included all the criteria applied. The participation of women from Eastern Poland was set at 51.3%, compared to 48.7% of men, while in Western Ukraine it was 51.7% and 48.3%, respectively. Residents of urban areas in Eastern Poland accounted for 47.5% of respondents, while in Western Ukraine it was 52.7%. Among Polish respondents, 7.8% had basic and vocational education level, 42.4% had secondary education, and 49.8% had higher education. In the case of Ukrainian respondents, 7.1% had basic and vocational education, 40.5% had secondary education, while 52.4% had higher education. In terms of age within the researched sample, 21.6% of the respondents from Eastern Poland were aged 18 to 25, 31.3% from 26 to 40 years, 20.9% from 41 to 55 years, and 26.2% of respondents were over 56 years old. Residents of Western Ukraine aged 18 to 25 constituted 26.1% of the respondents, those from the age of 26 to 40 accounted for 28.2%, ages 41 to 55 constituted 21.1%, and aged 56 and higher were 24.6%.

### 2.3. Statistical Methods Used

The Statistica 10.1 PL program was used in statistical analysis, and within it discriminant function analysis was applied to determine which variables differentiated consumers from Eastern Poland and Western Ukraine.

Classification functions were used in the form of calculation of their coefficients, which were determined for each group. Prior to the analysis, the authors performed tests to examine whether variables were normally distributed. It was assumed that the matrices of variances were homogenous in groups. Slight deviations were not of a great importance, due to the large number of respondents in each group. We considered *p*-values < 0.05 to be statistically significant.

## 3. Results

In the case of consumers from Eastern Poland [25], the research sample was calculated from 3,456,183 of its adult consumers, whereas in the case of Western Ukraine it was from 3,134,021 adult consumers [26]. While distinguishing regional food products from conventional products, the consumers of Eastern Poland, to a significantly higher degree at *p* < 0.001, pointed to the importance of television (TV) commercials. This proves that this promotion measure is used to a greater extent in Eastern Poland than in Western Ukraine, and effectively influences the opinion of its inhabitants. The value of the classification function, which is the result of a discriminatory function, in the case of Eastern Poland amounted to 1.511, comparing to 1.110 among consumers of Western Ukraine, who, at significantly higher levels where *p* < 0.001, paid attention to a product’s acquisition at designated points in their areas of residence. This means that in Western Ukraine, access to the place of purchase of regional products is greater than in Eastern Poland. The value of the classification function was 1.728, comparing to 1.380 for consumers of Eastern Poland. Additionally, consumers of the surveyed regions of Ukraine (1.558) searched for a suitable certification mark for a purchased product at significantly higher levels (*p* = 0.010) than those living in the eastern provinces of Poland (1.469). This is due to the fact that there were cases of food fraud in Ukraine [27]. Product and supplier labeling arouses greater consumer confidence, and the product ceases to be anonymous. (Table 1). 

To the highest degree, both the consumers of Eastern Poland and Western Ukraine obtained information on regional food products from their friends or family. A significantly higher value of classification function was shown among Ukrainians (2.337), at *p* < 0.001, than Poles (2.166). Such large values of the classification function shown in both groups testify to the importance of this method of obtaining information, i.e., information provided in direct contact. The same applies to information obtained through TV. The consumers of Eastern Poland, to a significantly higher degree (*p* < 0.001), used television to obtain information, which confirms the influence of TV commercials on distinguishing regional from conventional food. The value of the classification function in their case amounted to 2.138, comparing to 1.729 among the consumers of Western Ukraine, who obtained higher function values in the case of getting information through folders and leaflets (1.844) and by using the internet (1.225) than in the case of the consumers of Eastern Poland, where the function values were 1.411 and 0.853, respectively. In both cases, the difference was significant at *p* < 0.001. Poles, more often than Ukrainians, at a significant difference (*p* < 0.001) received information on regional food as a result of exploring these products while travelling in areas of their production (1.487) and through food fairs (1.041). In the case of the Western Ukraine consumers, these figures were, respectively, 1.111 and 0.794. Such acquisition of information may indicate that in the case of Poland, production centers and fairs are more often organized, making the product more accessible (Table 2).

Considerably more often, at *p* < 0.001, “high product quality” was declared by the consumers of Western Ukraine (2.799) than the consumers of Eastern Poland (2.485). Additionally, Ukrainians appreciated traditional and natural ways of producing these products (1.371) to a significantly higher degree, at *p* < 0.001, and pointed to the products’ health benefits as well (1.920). In the case of Poles, the values of classification function for these factors reached levels of 1.120 and 1.729, respectively. This may also indicate that in Ukraine, there are more traditional forms of food production than in Poland, where there is a higher specialization in production. On the other hand, Polish respondents (0.689) paid attention to the original taste of regional food products at a significantly higher level (*p* < 0.001) than Ukrainians (0.398). Both groups regarded recommendations of friends or family at a similar level. The value of classification function in the case of Eastern Poland was 1.811, while for Ukraine it was 1.766 (Table 3).

Consumers of both countries indicated having opportunities for purchasing regional food products. At the same time, they complained about the limited number of points of sale in both surveyed areas. Ukrainians (3.805), at a significantly higher level (*p* < 0.001), made such declarations than did Poles (3.599). Additionally, the classification function reached high values for answers that such products were hardly available. The situation is the result of a higher price and better quality of regional products than conventional ones, since there is no mass production of regional food. In the case of this declaration, significantly higher values, at *p* < 0.001, were declared by consumers of Eastern Poland (2.583) than Western Ukraine (2.456). This may indicate that both food markets do not fully respond to customer demand. Ukrainians raised bigger concerns about the uncertainty of origin of regional food products. In their case, the value of the classification function was 2.559, while among Poles it was 2.155, at *p* < 0.001. Consumers of Eastern Poland declared that these are hard-to-reach products, at the level of 2.106, compared to 1.497 in Western Ukraine. Significant differences occurred at *p* < 0.001. The survey results confirm that the regional products market is not fully keeping pace with the needs of residents of either Eastern Poland or Western Ukraine (Table 4).

The respondents of both countries made purchases of regional products mostly in their area of residence. Such declarations among consumers of Western Ukraine amounted to 2.385, comparing to 2.216 in the case of consumers of Eastern Poland. The high value of the classification function was achieved regarding the statement that products were purchased at regional fairs. In the case of Ukrainians, the declared value amounted to 2.091, and was significantly higher, at *p* < 0.001, than among Poles, whose value of the classification function was 1.876. It is probable that the tradition and taste of such products is highly preferred by consumers. As studies show, when buying in large stores, or when shopping on the internet, such products are preferred at lower degrees. Purchasing in supermarkets and hypermarkets had an average function value of 1.604 among consumers from Eastern Poland and 1.501 from Western Ukraine, at a significant difference (*p* = 0.019). Ukrainians purchased regional food products in regional food stores (1.033) significantly more often (*p* < 0.001) than Polish respondents (0.693). Both surveyed groups declared the purchase of regional products through mail order sales at a similar level (Table 5).

## 4. Discussion

Attitude toward something is an antecedent of intention, but it is also the degree to which an individual has a favorable or unfavorable evaluation of the behavior toward any purchase situation [18]. The aim of this paper, which assumed determination of awareness consumer measures towards regional food products in two surveyed countries, resulted in a conclusion that to a greater extent, the consumers of Western Ukraine buy regional products at designated points in the immediate area of residence, usually being suggested by a product certification mark. The consumers of Eastern Poland to the greatest extent watched out for the proper certification and labeling of products, most often being suggested by TV advertising. The obtained results confirm the outcomes of studies carried out in six European countries: Norway, Belgium, France, Spain, Italy, and Poland, which indicated that consumers largely purchase food products at local marketplaces and are open to innovations related to food production. They notice the fact that a product was produced under organic conditions, purchasing it even when its price is higher than conventional products’. The place of origin of the product and its labeling is also important for the consumer, as well as the use of organic materials and labels that would indicate the geographical origin of raw materials [28]. According to Salejda et al. [29], consumers should strive to raise awareness of their products and their quality, using the internet and television programs. Radzymińska and Jakubowska [30], in their research on the perception of regional and local food conducted among young consumers, confirmed that consumers perceive the value of regional, local, and traditional products, but also drew attention to insufficient promotion. Short supply chains are just beginning to be noticed by the consumer, although the barrier may be a higher price, which is the effect of a limited amount of a given product on the market.

It should be noted that contemporary consumers make purchases with great awareness of nutritional, taste, and health benefits of food products [31,32,33]. They appreciate to a greater extent local food products as those with higher health benefits [30]. It is a common conviction that producing in the local environment makes a product more valuable [34,35,36].

The increased interest in regional food products is a manifestation of new food-related tendencies, and in particular, an interest in preserving behaviors and values stemming from the cultural heritage of both Poland and Ukraine [36]. Consumers of Eastern Poland named TV, next to their family and friends, as an influential means of obtaining information about regional products. In the case of Western Ukraine, friends, family, and folders were the largest providers of information. These are sources that have an impact on the purchase of regional products. The products are appreciated for their regional flavor, which brings up the flavors of childhood, being associated with the heritage of the region that the consumers come from [37,38,39].

In both groups of respondents, when purchasing regional food products, the products’ high quality was definitely the most important characteristic. Additionally, strong health benefits were significant, as well as traditional and natural ways of their production. A similar pattern of behavior while purchasing regional products appeared among German consumers [40], who noticed their high quality and nutritional values having a positive effect on health. These opinions concerned meat products especially. Also, in other regions of the world—for instance, in Indonesia—there are policies of food diversification that are focused on the consumption of regional food [41].

Respondents pointed to the high availability of regional products, however limited by a small number of points of sale. Such problems were also emphasized by Gracia et al. [42] and Lang et al. [43] in their studies. They stated that the availability of local products in Spain and the United States is limited by a large distance from manufacturer to consumer. However, this problem can be overcome by consumers, who desire to buy the product that suits them and to be sure of its authenticity.

It was shown that sale of regional products to the greatest extent is carried out in the region of production, which is undoubtedly connected with problems resulting from distribution. That might also stem from a small potential for manufacturing regional products, which are usually sold at fairs and markets in the immediate area. There is a tendency to take over distribution of regional products by supermarkets and hypermarkets. A reflection of this is the policy of Tesco, an English chain of hypermarkets, which offers regional products. However, they must meet all quality and food safety standards [44].

The results from available research indicate that the mail order sale of regional products is also developing, as well as the creation of specialized regional food stores. In response to customers’ expectations, restaurants and hotels have also expanded their offerings, and include organic and regional products [8,45].

The growing interest in regional food is a manifestation of new food trends, and is implied by the desire to expose the behaviors and values resulting from cultural heritage [39]. In addition to human health-related aspects, several authors have analyzed the extrinsic, hedonic, and taste-related/sensory attributes that are important in the purchasing process [46,47]. An important factor supporting the development of regional food should be its promotion, which can take place through food festivals that allow the building of a positive image of the place of origin of regional food [48]. Consumers of regional food are often elderly people, which indicates that it is necessary to differentiate the offer and create products addressed to younger consumers, to show them through promotional tools why it is worth paying attention to regional products. The research conducted by Barska and Wojciechowska-Solis [39] also pointed at the benefits for the supply side of regional and traditional products, which are worth mentioning in this paper. It is worth noting that producers with traditional products in their assortment structure, which consumers associate with the regional ones, can count on the following benefits:-increase in demand for such products;-increased consumer confidence;-possibility to sell at higher prices;-financial support from the European Union funds under the Common Agricultural Policy (Poland is currently using these funds, and should Ukraine join the European Union, it will also receive support);-protection of labeled products against adulteration and other practices that could mislead the consumer as to the origin of the product name or characteristics;-preservation of biodiversity in the agricultural production and culinary diversity of regional products;-positive effect of production and sales of regional products on the activation of the region’s economy.

## 5. Conclusions

In the studied border regions (Eastern Poland and Wester Ukraine), there is a need for decisive promotional steps, since acquiring knowledge from family and friends is an insufficient method for spreading awareness. TV and Internet, especially on a regional scale, have a great promotional potential to educate young generations of consumers interested in the purchase of regional food products.

The high awareness of the importance of regional products among consumers of both countries should lead to an increase in the number of points of sale.

## Figures and Tables

**Table 1 foods-08-00467-t001:** Ways of distinguishing regional food products from conventional products.

Ways	Wilks’ Lambda: 0.685 *F* = 38.134; *p* < 0.001 *			Country of Residence	
	Wilks’lambda	*F* Value	*p* Value	Poland *p* = 0.517	Ukraine *p* = 0.483
Based on TV commercials	0.653	103.402	0.001 *	1.511	1.110
I purchase at designated points in my region	0.639	24.610	0.001 *	1.380	1.728
I look for the right product certification mark	0.601	6.586	0.010 *	1.469	1.558
Constants				11.140	11.933

* level of significant difference at *p* < 0.050. Source: Authors’ own analysis based on study material.

**Table 2 foods-08-00467-t002:** Obtaining information on regional food products.

Methods of Obtaining Information	Wilks’ Lambda: 0.628 *F* = 65.762; *p* < 0.001*			Country of Residence	
	Wilks’ Lambda	*F* Value	*p* Value	Poland *p* = 0.517	Ukraine *p* = 0.483
From folders and flyers	0.532	137.567	0.001 *	1.411	1.844
I get to know while traveling in the areas of production of regional products	0.602	81.558	0.001 *	1.487	1.111
From television and radio	0.589	97.598	0.001 *	2.138	1.729
From the Internet	0.532	78.098	0.001 *	0.853	1.225
From food fairs	0.587	31.497	0.001 *	1.041	0.794
From friends/family	0.546	10.590	0.001 *	2.166	2.327
Constants				15.795	18.509

* level of significant difference at *p* < 0.050. Source: Authors’ own analysis based on study material.

**Table 3 foods-08-00467-t003:** Reasons for buying regional food products.

Type of Reason	Wilks’ Lambda: 0.621 *F* = 35.753; *p* < 0.001*			Country of Residence	
	Wilks’ Lambda	*F* Value	*p* Value	Poland *p* = 0.517	Ukraine *p* = 0.483
Traditional and natural method of production	0.621	60.281	0.001 *	1.120	1.371
High product quality	0.619	35.687	0.001 *	2.485	2.799
Natural taste and smell	0.589	47.182	0.001 *	0.689	0.398
High health benefits of the product	0.581	17.252	0.001 *	1.729	1.920
Recommendation of friends/family	0.586	1.606	0.206	1.818	1.766
Constants				15.179	17.270

* level of significant difference at *p* < 0.050. Source: Authors’ own analysis based on study material.

**Table 4 foods-08-00467-t004:** Assessment of the availability of food regional products.

Assessment of Availability	Wilks’ Lambda: 0.608 *F* = 66.160; *p* < 0.001*			Country of Residence	
	Wilks’ Lambda	*F* Value	*p* Value	Poland *p* = 0.517	Ukraine *p* = 0.483
These are hard-to-reach products	0.602	205.867	0.001 *	2.106	1.497
Uncertainty of their originality	0.587	72.802	0.001 *	2.155	2.559
They are available, but in a limited number of points of sale	0.583	14.592	0.001 *	3.599	3.805
They are easily available	0.589	9.752	0.001 *	2.583	2.456
Constants				16.240	16.337

* level of significant difference at *p* < 0.050. Source: Authors’ own analysis based on study material.

**Table 5 foods-08-00467-t005:** Places for purchasing regional food products.

Place of Purchase	Wilks’ Lambda: 0.532 *F* = 41.642; *p* < 0.001*			Country of Residence	
	Wilks’ Lambda	*F* Value	*p* Value	Poland *p* = 0.517	Ukraine *p* = 0.483
In specialized stores with regional food	0.521	72.845	0.001 *	0.693	1.033
Only in the region where food is produced	0.551	18.809	0.001 *	2.216	2.385
At regional bazaars and fairs	0.552	29.281	0.001 *	1.876	2.091
In supermarkets and hypermarkets	0.521	5.460	0.019 *	1.604	1.501
Online/mail order sales	0.537	1.357	0.244	1.198	1.245
Constants				13.029	15.531

* level of significant difference at *p* < 0.050. Source: Authors’ own analysis based on study material.

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
