# Peer review of "Consumer Awareness of the Regional Food Market: The Case of Eastern European Border Regions"

_foods, 2019, doi:10.3390/foods8100467_

Round 1

Reviewer 1 Report

Review-Consumer attitudes on the regional food market: the case of Eastern European border regions

The manuscript “Consumer attitudes on the regional food market: the case of Eastern European border regions” reports on a questionnaire carried out with >1,000 consumers in both western Ukraine and eastern Poland about regional foods. The abstract and introduction claim that the primary aim of the article is to document consumers’ attitudes towards regional foods, the manuscript does not actually do this. The closest it comes to documenting attitudes is a question about reasons for purchasing regional foods, which is not the same thing, particularly since respondents seem not to have been asked a question about reasons not to purchase regional foods. Attitudes would presumably capture both positive and negative views of regional foods, while the question that respondents were actually asked only provides an opportunity to respond about positive aspects of regional foods. The analysis is entirely devoted to identifying differences between consumers in the two regions (Ukraine, Poland) without considering other demographic characteristics that could influence responses, such as age, gender, education, etc, which might be interesting to know about. This focus on differences between Ukraine and Poland is also not clear from the title, abstract, or introduction, so that needs to be addressed. The authors need to include text explaining how to interpret the estimates that they report; currently, they just state the number and move on. Some readers may not be familiar with the particular technique they use and so will get less out of reading the paper than if the authors provided interpretation of estimates. As a last major point, some the claims made in the discussion are a stretch and are not supported by the findings reported in the manuscript. I highlight instances of that in specific comments below. Finally, I would encourage the authors to get an English-language editor to help improve the clarity and flow of the writing. I’ve tried to provide a few suggestions about ways to improve the writing, but do not have time to thoroughly copy edit the manuscript.

Abstract

General: The abstract begins by saying that the aim of the paper is to determine the attitude of consumers…towards regional food products. However, the results discussed in the abstract only focus on where people get information about or can purchase regional food products, which is a different question. I think there is detail in the abstract that is unnecessary (talking about sample size calculation without reporting the sample size that is needed), which could be replaced with discussion of the results related to the motivating question of the article.

Line 12 (L12): There is a space missing between a period and the next word (“…products.A method…”). This a problem elsewhere in the abstract.

L12: The authors could eliminate the text “A method of…” and start with “A diagnostic survey…”. A couple of questions here: What is an author’s questionnaire? How was the survey diagnostic (I have not seen the term “diagnostic survey” used in social science research previously)? If the authors do not want to explain in what sense the survey was diagnostic here, perhaps they can eliminate the word “diagnostic” in the abstract and explain in greater detail in the body of the paper.

L16: I don’t think detail about sample size calculations are needed in the abstract.

L17: The authors can just say “Discriminant function analysis”—eliminating “The.”

L22: This should just be “young consumers.”

Introduction

L27: What does “food control” mean? This is used as disciplinary jargon in some fields—but the precise sense in which the authors mean it may not be understood by all interested readers. I encourage the authors to define what they mean by “food control.”

L28: I wonder if reference [2] is a mistake. I’ve looked at the article cited and it looks unrelated to the topic of the article.

L34: I recommend deleting “It is pointed out that” so that the sentence reads “The sector of agri-food producers is constantly…”

L36: This should be “…pay a higher price…” rather  than “…pay higher price….”

L45: The authors’ references to local food movements in the US are very dated—from 2003 and 2004—and one of them is, as far as I can tell, not actually a program (the “Local food purchase policy” quote). I would suggest looking up literature related to Farm to Table, Farm to School, farmers markets, community-supported agriculture (CSAs), or Buy Fresh Buy Local for a more accurate reference.

L52: There seems to be overlap in the goals the authors state here: “The effort was made to determine the reasons for buying regional food and the ways to distinguish it from conventional products. The availability of purchasing regional products was also assessed. The aim was also to identify the ways of distinguishing regional from conventional products as well as determining the place of its purchase.” Distinguishing/identifying regional foods is mentioned in both sentences, as is “availability/determining place of purchase.”

Materials and Methods

General question: How was the sample of respondents recruited? The authors do not note how they recruited participants.

L57: Again, what is an author’s questionnaire? Does this just mean a questionnaire developed by the authors? How was the survey diagnostic (I have not seen the term “diagnostic survey” used in social science research previously)?

L57: Are the questions provided for readers/reviewers? There seems to be no appendix, and the authors do not list a table where the questions are provided.

L59: What were the construction/validation procedures?

L64: What sample size did the sample size calculation indicate was needed?

L69-70: What do 3456183 and 3134021 represent? Are these the adult populations of consumers in eastern Poland and Western Ukraine? Were all adults considered to be consumers? Also, it’s typical in English to separate numbers with more than 4 digits using commas (e.g., 3,456,184 and 3,134,021).

L71: Many would consider the text in this section to be more appropriately reported in the “Results” section of the paper.

L72: This repeats the first sentence of the previous section word-for-word through the first 11 words of the sentence. In addition to addressing the repetitiveness, I also believe the sentence would read better if the latter part of the sentence was re-written to say “to examine 1128 respondents from the Podlaskie, Lublin, and Subcarpathia regions of eastern Poland and 1072 from the Volyn, Lviv, and Transcarpathia regions of western Ukraine.”

L76: How was lack of reliability and errors assessed by the authors to determine that a survey should be discarded?

L88: What does “the emerging groups” mean? Where these groups defined by the authors or by statistical analysis?

L89: It seems that the sentence “Classification functions were used in the form of calculation of their coefficients which were determined for each group” may not be written correctly—for instance, whose coefficients does the sentence refer to?

L90: This should likely be rewritten to say that prior to analysis, authors performed tests to examine whether variables were normally distributed.

L93: I would rewrite this to say “We considered p-values < 0.05 to be statistically significant.” The way in which it is currently written is convoluted (and not entirely correct).

Results

L98: What does a value of a classification function equal to 1.511 vs. 1.110 mean? How does one interpret that?

L105: It’s not necessary to both report a p-value and include an asterisk to denote significance at a specific level (e.g., * = P<0.05) since you already get that information from the p-value. “Constant” is also misspelled in the table.

L121: The authors have not Anglicized the names of the countries in Table 2.

L124: “Considerably more often, at P<0.001, such reason was declared by the consumers of Western Ukraine than the consumers of Eastern Poland.” What reason are the authors referring to hear?

L137: The authors use commas instead of periods to denote fractions of a number. What does “original taste and smell” mean? Is it supposed to refer to products having traditional characteristics for the region?

L140: The results reported in this paragraph seem contradictory. Polish consumers are more likely to both say that regional food products are hard to reach and to say that they are easily available, while Ukrainian consumers are more likely to say that they are available but in a limited number of points of sale. What do the authors make of these results?

Discussion

L169: In the introduction, the authors state that the aim of the paper is to “…identify attitudes of consumers of Eastern Poland and Western Ukraine towards regional food products.” Based on my reading, I do not see any study of the attitudes of these consumers towards regional food products. The authors study where they get information to distinguish regional from conventional products, where they get information on the products, reasons for buying products (which is not the same thing as attitudes towards the products), perceptions of the availability of the products and where they purchase them. All of the analyses are aimed at distinguishing between consumers from eastern Poland and western Ukraine, rather than reporting pooled results (which is what the stated aim in the introduction would suggest).

L171: Where does it ask about purchasing products in the immediate area of residence? Are they referring to the “only in the region where food is produced” item? Couldn’t this also be people traveling through an area, or is there additional information that isn’t reported in the table?

L174: Based on the evidence reported in this paper, I don’t think the authors can say that it confirms results from these other studies, because the authors don’t report evidence showing that consumers in Poland/Ukraine “largely” (does that mean a majority?) purchase food products in local marketplaces. [Side question: do the authors mean largely purchase regional food products in local marketplaces, or any food products?] The results say nothing about consumers in Poland/Ukraine being open to innovations related to food production. If the authors have evidence suggesting that, they do not report it here.

L177: Who notices the fact that it was produced organically and purchased it even when it costs more? There is no way that these consumers have driven conventional products out of the marketplace entirely because the only select higher priced organic products.

L181: All of the articles cited to support the claim that contemporary consumers make purchases with great awareness of nutritional, taste and health benefits are about organic or local food networks. Consumers who are active in those sorts of networks are likely unrepresentative of the general population—if everyone were acting with great awareness of nutritional and health benefits of foods, we would not likely be seeing the increases in obesity across all sorts of countries that we are. It may be true that local foods consumers are more aware of those attributes, but the authors would need to cite other articles—ones not focused on local/organic food networks—to make the broader claim.

L182: “They appreciate to a greater extent local food products as those with higher health benefits.” This needs a citation to support the claim.

L222: Many (if not all) of the items the authors list as benefits for producers from production of regional foods are not addressed by the results presented by the authors. Where does evidence of the accuracy of these statements come from, since it doesn’t come from survey data?

L241: The authors failed to delete the “Appendix B” text from the template.

Author Response

Review 1

At the beginning I would like to thank you for valuable tips in order to improve the article.

Below I provide answers to suggestions for improvement or questions in lines. In the current version of the paper, there will be inconsistencies in the lines, because it was supplemented with the proposed tips of the reviewer. All changes in the text are marked.

The abstract has been supplemented with additional information and unnecessary details suggested by the reviewer have been removed.

Line 12 (L12): Punctuation has been corrected.

It has been included what is "diagnostic test".

L.16 Calculations, information on sample size in the regions examined has been removed.

L.17: Corrected.

22: Revised according to the reviewer's instructions.

Introduction

L.27: A development of the term 'food control' has been posted.

L.28: References corrected, sorry for the error.

L.34: Word deleted.

L.36: Corrected, completed with "a".

L.45: The information has been supplemented with Farm to Table. Supplementing the reference.

L.52: The reviewer's suggestions have been taken into account and corrections have been made.

Materials and methods

Information on the sample has been supplemented in the article.

L.57a: The survey questionnaire was developed by the authors for the purposes of the study.

L.57b: Supplemented in the text. The research was conducted using computer-assisted telephone interviewing technique.

L.59: Information is provided in the text.

64: Before the start of the study, the authors calculated the sample based on census data so that the regions could be compared and the sample was representative.

L.69-70: Fixed numbers have been separated by commas.

L.71: Suggested text has been moved to the "Results" section.

L.72: The text has been corrected as recommended by the reviewer.

L.76: Deficiencies or errors in filling out decided the rejection of the survey. E.g. no answer to the question or giving a double answer.

L.88: 'Emerging group' were defined in the survey questionnaire. For a given study, it is a group of Polish and Ukrainian residents.

L.89: The text was given for linguistic correction. The sentence was checked by a native speaker.

L.90: The sentence was corrected in accordance with the reviewer's recommendations.

L.93: The sentence has been corrected.

Results

L.98: Information on what the classification function is is in the text

L.105: Thank you for the proposal to remove the "*" sign, but I have included references in the references regarding the use of discriminant analysis, where the description contains the "*" sign.

L.121: Names have been corrected.

L.124: The reason is given in the text.

L.137: Fixed.

L.140: I apologize for the error that crept into the interpretation of the results.

Discussion

L.169: Information was supplemented in the discussion.

L.171: Residents of the mentioned regions took part in the study.

L.174: For the countries surveyed, regional products are in a sense innovative products. Recently, regional markets and places where you can buy regional products are returning to popularity, food production was rather massive.

L.177: I agree with the reviewer that there is no question of crowding out conventional products from the food market, they will always be competitive compared to regional or traditional and ecological ones. But now consumers are more aware of what they are eating.

181: Literature has been supplemented.

L.182: Literature has been supplemented.

L.222: Information has been completed. The statement comes from the co-author's own research.

L.241. "Annex B" - Has been deleted.

Reviewer 2 Report

The manuscript attempts to clarify the question of how consumers in Western Ukraine and Poland distinguish regional from conventional food and where they buy it. The authors conducted an extensive survey in both regions.

Thank you for the opportunity to review the manuscript: The text is well written, the argumentation pragmatic but purposeful and well structured, and the question fundamentally interesting (although I will note in the critique that it is not clear to me why especially Poland and Western Ukraine were examined).

From my point of view, the greatest strength of the text is the methodical approach to the question. It is interesting and useful. The results provide an insight into the answer to the research question.

Nevertheless, I have two essential concerns: Firstly, the survey method is described far too sparsely. How exactly, for example, were the attitudes measured? Which scale was used - and how was it developed? Or: Why were only the ways to distuingish regional foods from conventional products described examined and no others? At the first glance, I miss social media and other advertising channels than TV. In addition: In which language was the survey conducted? How was it ensured that the questions were understood similarly in both regions? Behind these questions are not so much methodological concerns but rather the request to the authors to provide more information in order to be able to assess the quality of the results and the limitations of the study.

There is a second point associated with this that I think could be improved: I lack a very general explanation as to why these two countries in particular are being compared (in fact the results are mainly about comparison and not absolute values). Here, too, I do not deny that there are good reasons for selecting these two countries. Unfortunately, the manuscript provides no arguments on this point.

Author Response

Review 2

Thank you very much for the valuable tips and questions of the reviewer, which helped improve the quality of the paper.

Answering the question why the authors took care of this region, it should be said that despite the official state border there is no barrier to cross the border. There is visa-free travel. The research sample was selected in such a way that after statistical analysis it was possible to compare these two regions. The difference is that Poland is a member of the European Union and Ukraine applies to EU structures.

The "social media" mentioned by the reviewer were included in the questionnaire, unfortunately the received values did not qualify this variable to be included in the classification function model. Information on the Likert scale determination and information about the language in which the research was conducted was included in the text. Residents of Poland received a questionnaire in Polish, Ukrainians in Ukrainian.

References has been supplemented.

Round 2

Reviewer 1 Report

The authors have made some improvements to the manuscript “Consumer attitudes on the regional food market: the case of Eastern European border regions,” which reports on a questionnaire carried out with >1,000 consumers in both western Ukraine and eastern Poland about regional foods. There is a continued focus in the abstract and introduction that a primary aim of the article is to document consumers’ attitudes towards regional foods. However, the manuscript does not have anything to say about attitudes—at least according to the definition of “attitude” that I use. If the authors have a specific definition of “attitude” in mind that they feel incorporates the questions they asked in the survey, they should provide this definition and state a supporting document (e.g., dictionary).

A note about the authors’ response to my previous review: I would greatly appreciate it if my original comments are included just before your response to them. I can—and will—go back and look up my exact comment when reading the authors’ responses, but it would be much quicker if I can read the authors’ answers directly below the question that generated the response.

Abstract

L13-15: “Attitudes are cognitions and can sometimes be directed towards the brand, which is the product regional designation. So it is necessary to comment that attitudes towards regionality can generate a behavioral intent.” My issue with the previous text in the abstract was that “attitudes” were not being measured directly. This added text doesn’t eliminate that concern. The measures collected in the survey are not clearly related to attitudes. I would rather have the authors clearly state what they do measure than make it look like they collect measures of attitudes and then not. Based on the paper, it seems as though the authors collect information about:

Ways of distinguishing regional food products from conventional food products How people obtain information on regional food products Reasons for buying regional food products Assessment of the availability of regional food products Places to purchase regional food products

To me, these are measures of awareness, attributes that increase an individual’s intention to purchase, and perceptions of regional food products. Attitudes would be some statement(s) about how respondents feel about regional food products.

L16: I would just say “A diagnostic survey…” The text before that in the sentence isn’t necessary.

Introduction

L42: “In many countries, the support has emerged…”

L45: “Purchasing of local goods can be motivated by…” Delete “of.”

L47: Start with “Agri-food producers are constantly looking…” Delete “The sector of…”

L55: “It also affects elements such as information…” What does “it” refer to here? It is not clear from the text.

L76-77: “Poland is a member of the European Union, Ukraine applies to enter the EU structures.” I would change to “Poland is a member of the European Union (EU), while Ukraine has applied for EU membership.”

L77-78: “Despite the proximity of the two regions, there are significant differences in the functioning of the food markets in these two regions, as well as consumer awareness.” I would recommend revising the sentence to read, “Despite the proximity of these two regions, there are significant differences in the functioning of their food markets, as well as differences in consumer awareness.” This last point (differences in consumer awareness) needs a citation to support the assertion, however.

L108-9: “The sample was recruited among persons declaring the purchase of regional products.” I would change this to “The sample comprised individuals who reported purchasing regional products.”

L125: I think it would be better to say “…was applied to determine which variables differentiated consumers from eastern Poland and western Ukraine.”

L127: There needs to be a “to” between “Prior” and “the.”

L138-140: “(the classification function is the result of a discriminatory function, the higher the value of the classification function, the greater the importance of this factor).” The explanation of results is important enough that text explaining the results should be its own sentence. If the results reported are directly interpretable in some sense, that would be nice to know and the interpretation should be given. If they are not directly interpretable, it would also be helpful to know that.

L146: “This is due to the fact that there were cases of food fraud in Ukraine.” Do the authors know for certain that this is the cause of this finding? There are other instances of the authors making statement that explain some observed finding with certainty, which I think is not possible to definitively determine.

Author Response

Dear Reviewer,
Thank you very much for the next tips that helped improve the manuscript.
I copied the comments to the answer window and put my answers after the questions. As you asked.

Abstract

L13-15: “Attitudes are cognitions and can sometimes be directed towards the brand, which is the product regional designation. So it is necessary to comment that attitudes towards regionality can generate a behavioral intent.” My issue with the previous text in the abstract was that “attitudes” were not being measured directly. This added text doesn’t eliminate that concern. The measures collected in the survey are not clearly related to attitudes. I would rather have the authors clearly state what they do measure than make it look like they collect measures of attitudes and then not. Based on the paper, it seems as though the authors collect information about:

Ways of distinguishing regional food products from conventional food products How people obtain information on regional food products Reasons for buying regional food products Assessment of the availability of regional food products Places to purchase regional food products

To me, these are measures of awareness, attributes that increase an individual’s intention to purchase, and perceptions of regional food products. Attitudes would be some statement(s) about how respondents feel about regional food products.

As suggested by "attitudes", the term was replaced in the title and abstract with "consumer awareness measures". Changes to this term were made in the manuscript.

L16: I would just say “A diagnostic survey…” The text before that in the sentence isn’t necessary.

Taken according to suggestions.

Introduction

L42: “In many countries, the support has emerged…”

Excuse me, an error has occurred because support for the production of local products already exists and works in EU countries. Corrected.

L45: “Purchasing of local goods can be motivated by…” Delete “of.”

"Of" deleted.

L47: Start with “Agri-food producers are constantly looking…” Delete “The sector of…”

Done.

L55: “It also affects elements such as information…” What does “it” refer to here? It is not clear from the text.

"It" replaced "regional food supply chains".

L76-77: “Poland is a member of the European Union, Ukraine applies to enter the EU structures.” I would change to “Poland is a member of the European Union (EU), while Ukraine has applied for EU membership.”

Corrected.

L77-78: “Despite the proximity of the two regions, there are significant differences in the functioning of the food markets in these two regions, as well as consumer awareness.” I would recommend revising the sentence to read, “Despite the proximity of these two regions, there are significant differences in the functioning of their food markets, as well as differences in consumer awareness.” This last point (differences in consumer awareness) needs a citation to support the assertion, however.

Improved and supplemented with citation on differences in shaping consumer awareness (author:Князик Ю).

L108-9: “The sample was recruited among persons declaring the purchase of regional products.” I would change this to “The sample comprised individuals who reported purchasing regional products.”

Improved.

L125: I think it would be better to say “…was applied to determine which variables differentiated consumers from eastern Poland and western Ukraine.”

Changed according to reviewer's suggestions.

L127: There needs to be a “to” between “Prior” and “the.”

supplemented with "to".

L138-140: “(the classification function is the result of a discriminatory function, the higher the value of the classification function, the greater the importance of this factor).” The explanation of results is important enough that text explaining the results should be its own sentence. If the results reported are directly interpretable in some sense, that would be nice to know and the interpretation should be given. If they are not directly interpretable, it would also be helpful to know that.

As suggested by the reviewer, the interpretation of each Table includes your own opinion on the results obtained.

L146: “This is due to the fact that there were cases of food fraud in Ukraine.” Do the authors know for certain that this is the cause of this finding? There are other instances of the authors making statement that explain some observed finding with certainty, which I think is not possible to definitively determine.

The information was obtained from a research report in Ukraine. The authors of the study cited [27] in the references.

Reviewer 2 Report

Thank you for the opportunity to review the manuscript again. In my opinion, the authors have done a good job in implementing the comments and suggestions and have extensively revised the manuscript. I would be happy to see the manuscript published. I wish the authors every success with their research!

Author Response

Dear Reviewer,
Thank you very much for your kind words. Thank you very much for all comments and suggestions that have helped us improve the paper. We also hope that our future works will meet with the same approval.